# Dietary Hempseed Decreases Femur Maximum Load in a Young Female C57BL/6 Mouse Model but Does Not Influence Bone Mineral Density or Micro-Architecture

**DOI:** 10.3390/nu14204224

**Published:** 2022-10-11

**Authors:** Chandler A. Sparks, Hailey M. Streff, Derrick W. Williams, Cynthia A. Blanton, Annette M. Gabaldón

**Affiliations:** 1Hackensack Meridian School of Medicine, Nutley, NJ 07110, USA; 2College of Veterinary Medicine and Biomedical Sciences, Colorado State University, Fort Collins, CO 80523, USA; 3Harvard Medical School, Boston, MA 02115, USA; 4Department of Nutrition and Dietetics, Idaho State University, Pocatello, ID 83209, USA; 5Department of Biology, Colorado State University—Pueblo, Pueblo, CO 81001, USA

**Keywords:** hempseed, bone, biomechanics, C57BL/6 mice, antinutrients

## Abstract

Numerous seed and seed extract diets have been investigated as a means of combating age-related bone loss, with many findings suggesting that the seeds/extracts confer positive effects on bone. Recently, there has been rising interest in the use of dietary hempseed in human and animal diets due to a perceived health benefit from the seed. Despite this, there has been a lack of research investigating the physiologic effects of dietary hempseed on bone. Previous studies have suggested that hempseed may enhance bone strength. However, a complete understanding of the effects of hempseed on bone mineralization, bone micro-architecture, and bone biomechanical properties is lacking. Using a young and developing female C57BL/6 mouse model, we aimed to fill these gaps in knowledge. From five to twenty-nine weeks of age, the mice were raised on either a control (0%), 50 g/kg (5%), or 150 g/kg (15%) hempseed diet (*n* = 8 per group). It was found that the diet did not influence the bone mineral density or micro-architecture of either the right femur or L5 vertebrae. Furthermore, it did not influence the stiffness, yield load, post-yield displacement, or work-to-fracture of the right femur. Interestingly, it reduced the maximum load of the right femur in the 15% hempseed group compared to the control group. This finding suggests that a hempseed-enriched diet provides no benefit to bone in young, developing C57BL/6 mice and may even reduce bone strength.

## 1. Introduction

Bone formation and resorption is a dynamic process that is in flux throughout an individual’s lifetime. In the early years of life, bone formation exceeds resorption as the skeletal system develops. In later life, however, resorption often exceeds formation and age-related bone loss ensues [1]. Age-related bone loss is widely recognized as problematic due to its characteristic reduction in bone mineral density (BMD), micro-architecture, and strength. Each of these characteristic features of age-related bone loss places an individual at increased risk for bone fracture. Such fractures can be painful and costly, may lead to long-term disability, and can even be fatal [1,2,3]. 

The problem of age-related bone loss has prompted a significant amount of bone research. Such research aims to slow resorption later in life or to increase bone formation earlier in life and, thus, provide more bone mass to combat age-related bone loss [1]. The link between diet and bone health is well established and, as a result, a significant amount of bone research aims to identify nutrition-related strategies to slow bone resorption and/or promote bone formation [4]. Various seeds and seed extracts have been previously investigated as dietary supplements with the potential to improve bone health due to their characteristic rich nutrient profile, high concentrations of polyunsaturated fatty acids, antioxidants, and anti-inflammatory compounds. Of the dietary interventions investigated, grape seed extract [5], pomegranate seed oil [6], broccoli seed extract [7], flaxseed oil [8], chia seeds [9], fenugreek seed extract [10], safflower seed [11], and *Camelina sativa* oil [12] have shown positive effects on BMD, bone micro-architecture, and/or bone strength in a variety of animal models.

Despite the increased inclusion of hempseed in both human and agricultural animal diets for improving health, there has been little research investigating the effects of dietary hempseed on bone [13]. Like other seeds, hempseed is rich in digestible nutrients, polyunsaturated fatty acids, antioxidants (α-, β-, γ-, and δ-tocopherol), and anti-inflammatory compounds (α- & γ-tocopherol) [14,15,16,17]. Conversely, hempseed contains antinutritional factors, including trypsin inhibitors and phytic acid, which impair the absorption of proteins and minerals, respectively [18]. Hempseed also contains cannabidiol (CBD) to varying degrees (0.32 to 25.55 μg/g) [19]. Currently, there is not a comprehensive understanding of the effects of CBD on bone. However, it has been found that CBD enhances the biomechanical properties (maximum load and work-to-fracture) of healing femurs in rats by enhancing collagen cross linking [20]. Furthermore, activation of the CB2 receptor, a receptor predominantly found in peripheral tissues (spleen, blood cells, and bone) that are activated by CBD, has been noted to regulate several pro-osteogenic functions [21,22]. Thus, it is possible for dietary hempseed to have either positive or negative effects on bone properties.

The goal of this study was to determine the impact of a hempseed-supplemented diet on bone mineralization, micro-architecture, and biomechanical properties, using a relatively young C57BL/6 female mouse model. The bones selected for analysis were representative of cortical-rich bone (femur) and trabecular-rich bone (lumbar vertebra; L5), which were imaged using dual energy x-ray absorptiometry (DXA) and microcomputed tomography. The mechanical properties of the femur bone were also analyzed using three-point mechanical bending testing. The effects of hempseed on a single biomechanical property (fracture force) have been previously investigated [23]. Specifically, it was found that the tibia bones of laying hens fed a diet supplemented with 6% and 9% hempseed had increased fracture force compared to a control group. This indicated that the diet improved bone strength in the laying hens but did not provide insight into the effects of the diet on BMD, bone micro-architecture, or biomechanical properties other than fracture force (e.g., stiffness, yield load, work-to-fracture, post-yield displacement, etc.). In this study, we aim to fill these gaps in knowledge regarding the effects of dietary hempseed on skeletal bone properties.

## 2. Materials and Methods

### 2.1. Experimental Design, Diet, and Animals

Twenty-four female C57BL/6 mice (Charles River Laboratories, Wilmington, MA, USA) were obtained at three weeks of age and fed a standard control diet for growing mice in the following two weeks. At four weeks of age, the mice were implanted with a subcutaneous identifying microchip in the subscapular region (UID Identification Solutions, Lake Villa, IL, USA). At five weeks of age, the mice were randomly divided into three diet groups: a control diet without hempseed (CON; *n* = 8), a diet supplemented with 50 g/kg (5%) hempseed (5-HS; *n* = 8), and a diet supplemented with 150 g/kg (15%) hempseed (15-HS; *n* = 8). The diets were prepared by Dyets Inc. (Bethlehem, PA, USA) in pelleted form and were approximately isocaloric and matched for macronutrients (Table 1). For hempseed diets, the AIN-93G base diet was supplemented with the whole ground, organic, toasted hempseed (CHII Naturally Pure Hemp-Naturally Splendid Enterprises, Ltd., Pitt Meadows, BC, Canada). Throughout the study, mice were fed their respective diets ad libitum and given water ad libitum. Fresh food was provided weekly to ensure no degradation of nutrients.

The mice were pair-housed in polycarbonate cages and kept on a 12 h:12 h light:dark cycle (lights on at 0600 and off at 1800). Mice were weighed weekly and assessed for health. All mice completed the study, and none developed adverse responses to the hempseed diet. At twenty-nine weeks of age, the mice were deeply anesthetized using isoflurane gas and euthanized via cervical dislocation. Their carcasses were then eviscerated and stored at −70 °C in phosphate-buffered saline (PBS) until the time of ex vivo experimentation to ensure that the skeletal bones were kept well hydrated. 

### 2.2. In Vivo DXA Scanning of the Femur and L5 Bone Mineralization

As the mice were raised under experimental conditions from five to twenty-nine weeks of age, their entire bodies (excluding the head of the mouse) were scanned using a Lunar PIXImus II DXA scanner (GE Healthcare, Chicago, IL, USA) at monthly intervals. The first scan was performed at five weeks of age and the last scan at twenty-nine weeks of age, for a total of seven scans per mouse throughout the study. On the day of scanning, mice were lightly anesthetized using 1.5–2% isoflurane gas to limit movement, then placed in a supine position on the imaging stage. Whole animal scans were obtained, and, from these images, individual bones were analyzed at a later time. To obtain the BMD values for specific bones, the region of interest (ROI) was adjusted to contain only the bone of interest. For analysis of the right femur, a cortical rich bone, the ROI was placed around the distal condyles, diaphysis, and a portion of the femoral neck (Figure 1A). The femoral head was excluded to avoid capturing pelvic bone at the hip socket. For analysis of the L5 vertebrae, a cancellous-rich bone, the ROI was placed around the entire L5 vertebra (Figure 1B). 

### 2.3. Ex Vivo Micro-Computed Tomography Evaluation of the Femur and L5 Micro-Architecture

The frozen carcasses of mice collected at the end of the study at age twenty-nine weeks were transported to the University of Utah, where they were scanned at the Preclinical Imaging Core Facility. Scans were performed using a Siemens Inveon Microcomputed Tomography Scanner (Siemens Medical Solutions, Malvern, PA, USA). The carcasses were kept hydrated in PBS and were scanned using step-and-shoot geometry with 540 projections, an X-ray voltage of 45 kV, a current of 300 mA, and an exposure time of 3000 ms. The effective pixel size of the scans was 19.57 microns.

All analyses of micro-computed tomography images were performed using NIH ImageJ software (Version 2.1.0/1.53c) with BoneJ plug-in [24]. Representative images showing the parameters analyzed for femur diaphysis and L5 centrum trabecular region are shown in Figure 2A–C. For the femur, the entire diaphysis length was first measured then divided by two to determine the exact mid-diaphysis site for analysis of cortical thickness (Ct.Th, mm) and cortical area fraction (Ct.Ar/Tt.Ar, %). The mid-diaphysis region was selected because this was the site where three-point mechanical strength bending tests would be subsequently performed. Unaltered microcomputed tomography images were used for Ct.Th analysis by measuring the linear distance (mm) between periosteal (outer) and endosteal (inner) surfaces, starting at the anterior face and then subsequently every 45º for each cross-sectional image (CSI) (*n* = 8 measurements per CSI). The eight measurements were then averaged for an individual CSI. This was repeated for a total of seven cross-sectional images at the mid-diaphysis region (the CSI at the exact mid-point, three immediately proximal to the exact mid-point, and three immediately distal to the exact mid-point), yielding the mean Ct.Th of the right femur mid-diaphysis region. The same seven cross-sectional images used for the analysis of Ct.Th were then converted to binary images for analysis of Ct.Th/Tt.Ar (Figure 2B). Incidentally, the right femur was incompletely captured in the scans of two mice (one mouse from CON and one from 15-HS) and could not be analyzed.

For the L5 vertebra, bone volume fraction (BV/TV), connectivity density (Conn.D), trabecular thickness (Tb.Th), and trabecular separation (Tb.Sp) were analyzed. The ROI only included the cancellous bone within the vertebral centrum (i.e., the main vertebral body). It excluded the cortical shell of the centrum and the vertebral processes (Figure 2C). This was done to ensure that only the trabecular structure was captured for analysis of trabecular bone parameters.

### 2.4. Three-Point Bending of the Femur

The mechanical strength of the femur mid-diaphysis was evaluated using a three-point bending tester (Figure 3A). The machine was constructed by the Colorado State University-Pueblo Engineering Department and has been used in previous studies on both rat and mouse bones [25,26]. Prior to testing, the right femur bone from each mouse carcass was dissected from the lower extremity and all soft tissues were removed. Using a pair of digital calipers, the length of the femur bone was measured, and the mid-point of the diaphysis was identified. The mid-point was then marked using a permanent ink black marker to give a target to lower the crosshead beam onto, thus ensuring that the load was applied to the mid-diaphysis region (Figure 3B). The bone was then placed into a PBS bath to ensure it was kept well hydrated before the test began (Figure 3A). 

To begin testing, the bone was placed anterior face upward onto two support beams with a span width of 10 mm (Figure 3B). The top surfaces of the support beams were covered with 80-grit sandpaper to ensure that the bone samples did not shift or rotate during testing. The machine utilized in the present study was equipped with a calibrated iLoad Mini force sensor, model MFM-010-050-S (LoadStar Sensors, Fremont, CA, USA) with 44.48 N (10 lb.) capacity and 1.0% accuracy, and a Mitutoyo displacement sensor, model ID-S112EX (Mitutoyo, Aurora, IL, USA) with 0.001 mm resolution and 0.00305 mm accuracy. The sensors were connected to a computer through a 24-bit load cell interface, model DQ-1000 (LoadStar Sensors, Fremont, CA, USA). An Extech Instruments external power source, model 382,213 (Extech Instruments, Nashua, NH, USA) was connected to the machine to supply power.

Before each test, it was ensured that the crosshead beam was lowered at a speed of ~1 mm/min (mean = 0.9985 mm/min; range = 0.991–1.004 mm/min). At the start of testing, the crosshead beam was lowered onto the anterior face of the bone at the point marked at the mid-diaphysis region (Figure 3B) until a pre-load force of ~0.5 N was established. The pre-load was important to establish before testing commenced to ensure no shifting or rotating of the bone would occur during the test. Baseline data, with the pre-load applied, was collected for five seconds as the crosshead beam sat still. The crosshead beam was then lowered onto the bone at a constant lowering speed (i.e., monotonic) and continued to be lowered until five seconds after bone fracture was observed. Data was logged into Excel in real-time using SensorVue software (LoadStar Sensors, Fremont, CA, USA).

### 2.5. Analysis of Load-Displacement Curves

In the present study, common extrinsic (i.e., whole bone) mechanical properties were analyzed. Thus, load-displacement curves were generated from the load (N) and displacement (mm) measured during the three-point bending test using IgorPro software (Version 8.04) (WaveMetrics, Portland, OR, USA). An example load-displacement curve is shown in Figure 4 and the accompanying table (Table 2) shows the parameters obtained from the graph.

### 2.6. Statistical Analysis

All statistical analysis was done in RStudio (Version 1.4.1106). Where data were a repeated measure with two independent variables, one a within-subjects variable and the other a between-subjects variable, a two-way mixed effect ANOVA with interaction was used to analyze the data. This test was specifically used for the analysis of DXA data, with age as the within-subjects variable and diet group as the between-subjects variable in the model. All other data, where there was one independent variable (diet group) and one parameter of interest, were analyzed using a one-way ANOVA. Where significance was found, a *Tukey–Kramer post hoc* test was used. Data was ensured to be approximately normal using a Shapiro–Wilk test when a one-way ANOVA was used. Where data was non-normal, per the Shapiro–Wilk test (work-to-fracture), a Kruskal-Wallis test was used. 

## 3. Results

### 3.1. Right Femur and L5 BMD

A two-way mixed effect ANOVA of the right femur BMD values obtained from DXA scanning revealed a significant main effect of age on BMD (*p* < 0.001). The same was true for the BMD of L5 (*p* < 0.001). No further investigation of these findings was performed, as it is unsurprising given that the mice were young and growing. There was no significant main effect of diet group on the BMD of the right femur (*p* = 0.31) or L5 (*p* = 0.40). The data of L5 did not show any interaction between age and diet group (*p* = 0.29) but the data from the right femur did show a significant interaction term (*p* = 0.016). Further investigation showed that the right femur BMD of mice in the 15-HS group did not increase between 17–21 (*p* = 0.99) and 21–25 (*p* = 0.059) weeks of age, whereas the right femur BMD of mice in the CON (17–21 weeks *p* = 0.049; 21–25 weeks *p* = 0.016) and 5-HS (17–21 weeks *p* = 0.005; 21–25 weeks *p* = 0.008) group did show an increase between these two time intervals. Interestingly, the right femur BMD of mice in the CON group decreased between 25–29 (*p* = 0.019) weeks of age and returned to approximately the same value as the other two groups. All results for right femur BMD and L5 BMD changes with age according to diet group are shown in Figure 5A,B, respectively.

### 3.2. Cortical and Trabecular Micro-Architecture

Regarding cortical structure, a one-way ANOVA revealed no significant difference in Ct.Th (*p* = 0.68) or Ct.Ar/Tt.Ar (*p* = 0.84) between the groups. Furthermore, there was no significant difference in the trabecular micro-architectural parameters between the groups, including BV/TV (*p* = 0.88), Conn.D (*p* = 0.88), Tb.Th (*p* = 0.72), or Tb.Sp (*p* = 0.71). These findings are summarized in Table 3.

### 3.3. Femur Biomechanical Properties 

Per a one-way ANOVA, the data from three-point bending showed no significant difference in the stiffness (*p* = 0.18), yield load (*p* = 0.24), or post-yield displacement (*p* = 0.58) between the groups. Due to a non-normal distribution of work-to-fracture amongst CON (W = 0.81, *p* = 0.040), 5-HS (W = 0.70, *p* = 0.002), and 15-HS (W = 0.71, *p* = 0.003), a Kruskal-Wallis test was used for analysis, which showed no significant difference in work-to-fracture between the groups (*p* = 0.69). One-way ANOVA did show a significant difference in maximum load between the diet groups (*p* = 0.026). A *Tukey–Kramer post hoc* analysis revealed that the maximum load of the right femora was significantly lower in the 15-HS group compared to the CON group (*p* = 0.030). There were no significant differences in the right femora maximum load between CON vs. 5-HS (*p* = 0.079) or 5-HS vs. 15-HS (*p* = 0.89) (Table 4, Figure 6). 

Despite a lack of significance, many parameters showed values that largely decreased from CON to 15-HS. For instance, stiffness was 18.63% lower in 15-HS vs. CON and yield load was 18.50% lower in 15-HS vs. CON. Again, however, neither of these parameters showed a *p*-value of <0.05 per one-way ANOVA.

## 4. Discussion

In the present study, young female C57BL/6 mice were raised from 5 to 29 weeks of age on diets supplemented with either 0%, 5%, or 15% whole hempseed. During this period, the BMD of the right femur and L5 vertebra of each mouse was measured at monthly intervals with DXA scanning. The femur bone was chosen as a cortical-rich bone and the L5 vertebra was chosen as a cancellous-rich bone, thus allowing for the analysis of dietary influences of hempseed on different bone types. For both bones, age showed a significant main effect on BMD. This finding is unsurprising given that the mice were young, growing, and expected to have bones with increasing BMD. The finding that the diet group did not have a significant main effect suggests that across age points, the diet did not seem to influence BMD. The finding that there was a significant interaction of age and diet for the right femur suggests that at some age points the BMD of the right femur is different between at least two groups. Specifically, the right femur of mice fed the 15% hempseed group did not accumulate significant BMD between 17–25 weeks of age (Figure 5). This finding could suggest that mice fed a 15% hempseed diet accumulate BMD to a lesser extent as compared to mice fed a 0% or 5% hempseed diet as they begin to enter mature adulthood (3–6 months). However, this trend did not continue between 25–29 weeks of age and by the last time point, BMD for all diet groups was approximately the same. Given this, it is difficult to conclude if the statistically significant finding is of any physiologic significance. An extension of the study beyond 29 weeks of age may reveal whether or not the 15% hempseed diet has a longer-term discernible impact on femur BMD.

Microcomputed tomography was used to determine the micro-architectural properties of the right femur mid-diaphysis and L5 centrum. The diet did not seem to significantly influence any of the micro-architectural parameters analyzed, which included indexes of integrity for both cortical and trabecular bone (Table 3). However, it should be noted that there are numerous other parameters for both cortical bone (e.g., cortical porosity, pore density, endocortical perimeter, etc.) and trabecular bone (e.g., bone surface density, structural model index, etc.) that were not analyzed in the present study [27].

Each biomechanical property analyzed in the present study reflects different qualities of the bone: e.g., stiffness (resistance to elastic deformation), yield load (strength), maximum load (strength), post-yield displacement (ductility), and work-to-fracture (toughness). Of the parameters analyzed, only the maximum load was significantly different between groups with the 15-HS group showing a lower mean maximum load compared to the control group (Figure 5). This finding suggests that a 15% hempseed diet may result in a reduction in bone strength. It should also be noted that there were some interesting trends without statistical significance (18.63% reduction in stiffness between 15-HS vs. CON; 18.50% reduction in yield load between 15-HS vs. CON). It is possible that a higher supplementation dose of hempseed, larger sample size, and/or extended length of time on the diet (>29 weeks of age) would reveal if the diet significantly reduces these parameters.

It is interesting that the hempseed diet reduced femur maximum load but did not seem to change BMD or bone micro-architecture, given that BMD accounts for ~60% of bone strength and that bone micro-architecture is another strong factor underlying bone strength [28]. However, there are other contributing factors that were not investigated in the present study. For instance, the elemental composition of bone has been suggested to play an important role in bone health, can be used as an indicator of bone health, and can be determined from inductively coupled plasma mass spectrometry (ICP-MS) [29]. Furthermore, the composition of bone is not entirely inorganic but also includes collagen fibrils, cells, and numerous proteins that can influence the biomechanical properties of bone [1]. Structural analysis of collagen fibrils and histomorphologic analysis of bone are methods commonly performed that may reveal possible changes to the organic and/or cellular component of bone that may have contributed to our findings. In a separate study using the humerus bone from the same mice used here, we found that osteoblast density at the distal end plate region was reduced in both the 5-HS and 15-HS groups vs. the CON diet group [30]. This finding suggests that the hempseed diet may in fact promote some alterations in the cellular component of bone.

The finding that a hempseed-enriched diet reduced maximum load is also surprising given a few other considerations. First, Skřivan et al. [23] showed that the tibia of laying hens fed a diet supplemented with 6% and 9% hempseed had an increased fracture load compared to laying hens fed a control diet without hempseed. This finding indicates that hempseed supplemented in the diet at 6% and 9% has the potential to increase bone strength. In our study, however, the femur bones of mice fed a diet supplemented with a lower dose of hempseed at 5% showed no increase or decrease in strength parameters. Furthermore, at a higher dose of 15% the hempseed diet reduced a strength parameter of the femur bones in the mice. However, it should be acknowledged that the hempseed doses (5% vs. 6% & 15% vs. 9%) and the animal models were different between the study noted and the present study. Notably, the bones of laying hens undergo significant loss of BMD, cortical micro-architecture, and trabecular micro-architecture at the age of those used by Skřivan et al. (52-week-old), whereas young and still developing mice were used in the preset study [23,31]. 

The second reason these findings are surprising is that many other seed/seed extracts that are rich in nutrients, antioxidants, and anti-inflammatory compounds have shown positive influences on BMD, bone micro-architecture, and/or bone biomechanical properties. Our findings suggest that hempseed, though also possessing these qualities, may have a negative influence on bone strength. Though hempseed contains several digestible nutrients, antioxidants, and anti-inflammatory compounds, it also contains anti-nutrients. Specifically, it contains trypsin inhibitors and phytic acid [14,18]. 

Regarding bone health, phytic acid may be of especial importance given that it chelates minerals (iron, calcium, magnesium, zinc) and hinders their absorption across the gastrointestinal tract. It has been suggested that the phytic acid in hempseed may lead to mineral deficiencies over a long period of diet administration, as has been observed in other high-phytic acid foods that produce rickets (i.e., the softening and weakening of bones in children) such as oatmeal [18,32]. It has been shown that the addition of extra calcium as well as the addition of commercial phytin (CaMg phytate) to the diet antagonizes the rachitogenic action of phytic acid in cereals [32]. It has also been noted and suggested that different varieties of hempseed have different concentrations of phytic acid and that the phytic acid contents in hempseed meal should be reduced to avoid mineral deficiencies [18]. The previous investigation by Skřivan et al. had approximately equal amounts of calcium in the diets of the control and all experimental groups [23]. Furthermore, information about enzyme supplementation (e.g., phytase) to overcome the effects of antinutrients is not mentioned [23]. Given this, it is still unclear what effects dietary hempseed may have on bone when the actions of phytic acid are reduced. Future studies investigating hempseed diets with reduced phytic acid content, extra calcium, phytase enzyme supplementation, and/or added commercial phytin may reveal if a hempseed diet can be consumed without a reduction in bone strength and, perhaps, also, confer a positive effect on bone due to the other beneficial compounds in the seed when the actions phytic acid are controlled.

## 5. Conclusions

Our findings demonstrate that long-term dietary supplementation with hempseed in a relatively young and developing female C57BL/6 mouse model did not provide beneficial effects to the mechanical properties of the right femur. In fact, the diet may impact the strength of bone in a negative manner. Furthermore, the diet did not seem to influence the BMD (femur & L5), cortical micro-architecture (femur), trabecular micro-architecture (L5), and other extrinsic mechanical properties of bones (femur) in the mice.

## Figures and Tables

**Figure 1 nutrients-14-04224-f001:**
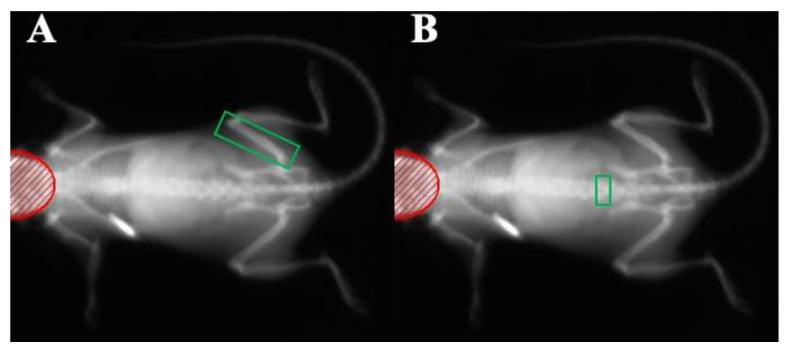
Representative DXA scanning images of a C57BL/6 mouse. (**A**) A scan with the ROI placed around the distal condyles, diaphysis, and a portion of the femoral neck (femoral head and pelvis excluded from ROI), and (**B**) the same scan with the ROI placed around the L5 vertebra. The region of interest (ROI) is shown in green. The bright white object at the left scapular region is the implanted subcutaneous microchip used for animal identification.

**Figure 2 nutrients-14-04224-f002:**
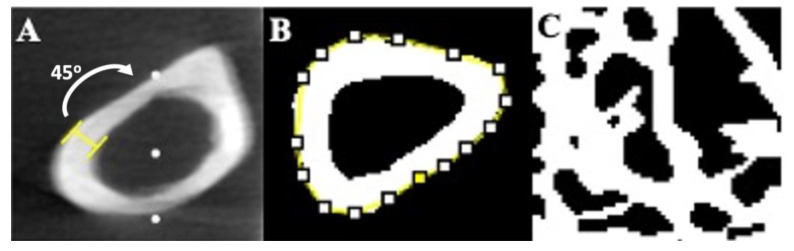
Representative micro-computed tomography images of femur and L5 centrum. (**A**) Femur Ct.Th measured at the mid-diaphysis as the distance between periosteal (outer) and endosteal (inner) bone surfaces using unaltered images. (**B**) Femur Ct.Ar/Tt.Ar values were obtained by making the image binary, where all bone material is made white and non-bone material is made black. An ROI (yellow) is then placed around the periosteum and Ct.Ar/Tt.Ar is measured as the fraction of white pixel to black pixel. This is repeated for every cross-sectional image within the mid-diaphysis region. (**C**) All trabecular bone measurements (BV/TV, Conn.D, Tb.Th, Tb.Sp) were obtained by isolating a square ROI containing only cancellous bone within the L5 centrum and making the image binary.

**Figure 3 nutrients-14-04224-f003:**
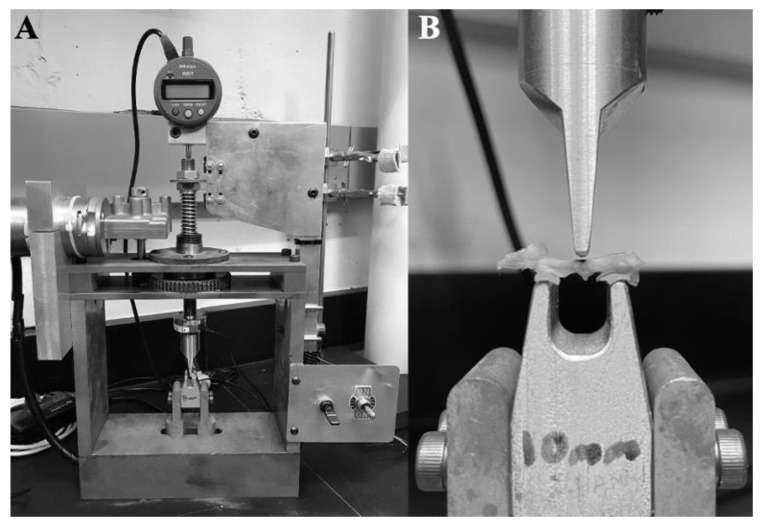
The three-point bender was used in the present study. (**A**) The full machine shown without a bone on the support beams. (**B**) The 10 mm support beams are shown with an example right femur bone, marked at the mid-point (black).

**Figure 4 nutrients-14-04224-f004:**
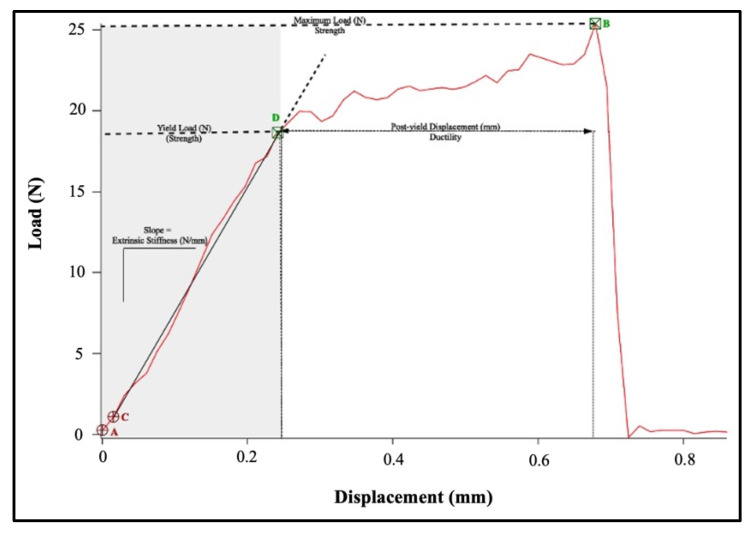
An example load-displacement curve in IgorPro. The entire curve from the beginning of the test to the fracture point is bound by the cursors A,B. Cursor B specifically marks the maximum load. Cursors C,D show the portion of the linear elastic region used to measure stiffness. The entire linear elastic region is shaded in grey. Cursor D also marks the yield point. All area not shaded in grey, past the yield point, is the plastic region.

**Figure 5 nutrients-14-04224-f005:**
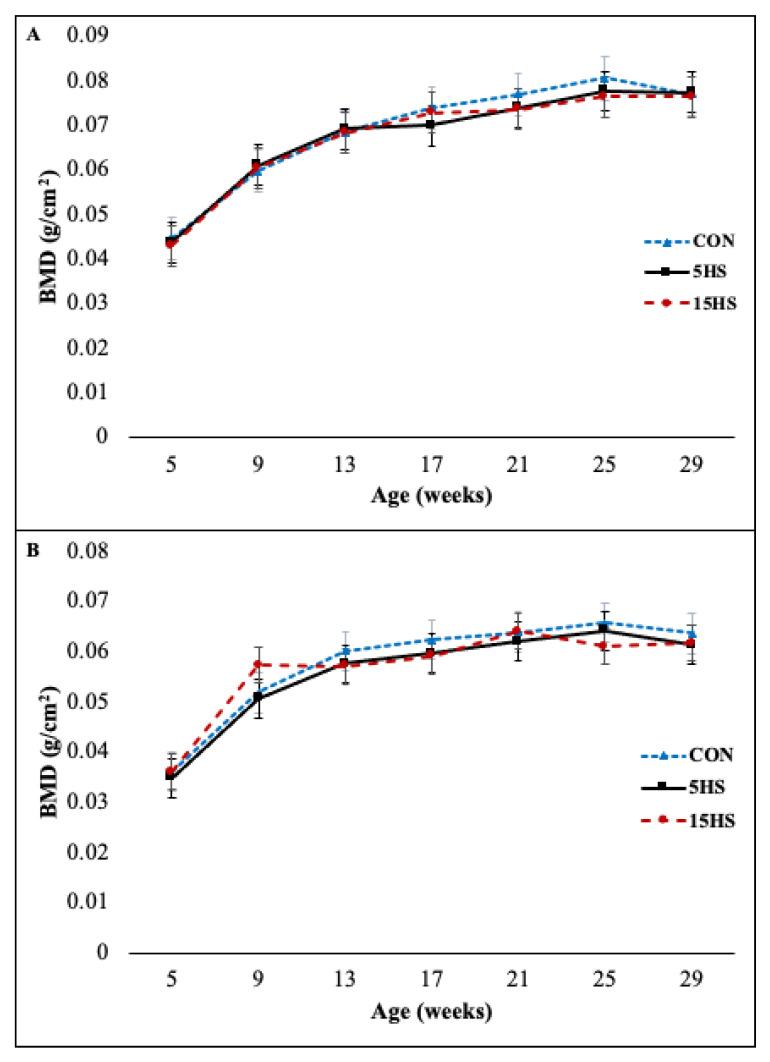
BMD of the right femur (**A**) and L5 vertebrae (**B**) with respect to age and diet group. Points represent mean BMD within a diet group and bars represent SEM of BMD within a diet group.

**Figure 6 nutrients-14-04224-f006:**
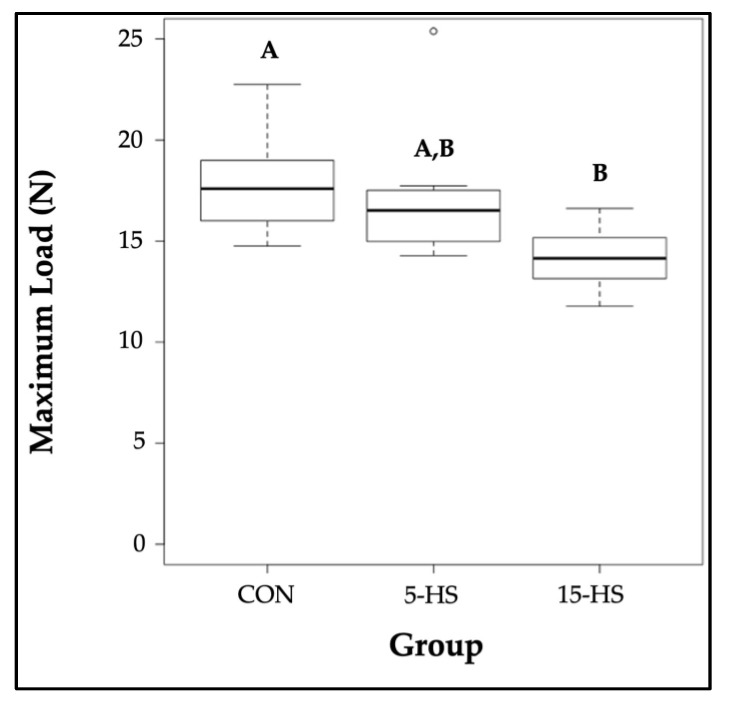
Boxplots of maximum load (N) of the right femur between diet groups. CON (median = 17.59, IQR = 2.60, range = 14.75–22.74), 5-HS (median = 16.52, IQR = 2.23, range = 14.28–25.38), and 15-HS (median = 14.14, IQR = 1.75, range = 11.78–16.62). Groups sharing the same letters above bars are not significantly different (*p* > 0.05) whereas groups with different superscripts are significantly different (*p* < 0.05) per the *Tukey–Kramer* test.

**Table 1 nutrients-14-04224-t001:** Experimental diet ingredients (g/kg).

Ingredient	Control Diet ^a^	5% Hempseed Supplemented Diet	15% Hempseed Supplemented Diet
Casein, High Nitrogen	200	185	155
L-Cystine	3	3	3
Sucrose ^b^	100	100	100
Cornstarch	397.486	395.99	392.997
Dyetrose	132	132	132
Soybean Oil	70	52	16
t-Butylhydroquinone	0.014	0.01	0.003
Cellulose	50	34.5	3.5
Mineral Mix #210025 ^c^	35	35	35
Vitamin Mix #310025 ^d^	10	10	10
Choline Bitartrate	2.5	2.5	2.5
Hempseed ^e^	0	50	150
Total	1000	1000	1000
Kilocalories per kg	3760	3814	3922

^a^ AIN-93G (Dyets Inc., Bethlehem, PA, USA); ^b^ Ninety percent tetrasaccharides and higher; ^c^ Composition Mineral Mix (g/kg): CaCO_3_ (357.0), KH_2_PO_4_ (196.0), K Citrate•H_2_O (70.78), NaCl (74.0), K_2_SO_4_ (46.6), MgO (24.3), Fe citrate (6.06), ZnCO_3_ (1.65), MnCO_3_ (0.63), CuCO_3_ (0.31), KIO_3_ (0.01), Na_2_SeO_4_ (0.01025), (NH_4_)_6_Mo_7_O_24_•4 H_2_O (0.00795), Na_2_SiO_3_•9 H_2_O (1.45), CrK(SO_4_)_2_•12 H_2_O (0.275), LiCl (0.0174), H_3_BO_3_ (0.0815), NaF (0.0635), 2NiCO_3_•3 Ni(OH)_2_•4 H_2_O (0.0318), NH_4_VO_3_ (0.0066); ^d^ Composition Vitamin Mix (g/kg): thiamin HCl (0.6), riboflavin (0.60), pyridoxine HCl (0.7), nicotinic acid (3.0), Ca pantothenate (1.6), folic acid (0.2), D-biotin (0.02), vitamin B12 (0.1% in mannitol) (2.5), vitamin A palmitate (500,000 IU/g) (0.8), DL-α-tocopherol acetate (500 IU/g) (15), vitamin D3 (400,000 IU/g) (0.25), vitamin K/dextrose 10 mg/g (phylloquinone) (7.5); ^e^ Whole ground, organic, toasted hempseed (CHII Naturally Pure Hemp- Naturally Splendid Enterprises, Ltd., Pitt Meadows, BC, Canada).

**Table 2 nutrients-14-04224-t002:** Whole-bone biomechanical properties measured in the present study and their procedures of measurement with respect to Figure 4.

Parameter	Cursors	Procedure
Stiffness (N/mm)	C–D	Slope between two points
Yield Load (N)	D	Load at point
Post-yield Displacement (mm)	D–B	Displacement between two points
Maximum Load (N)	B	Load at point
Work-to-Fracture (N⋅mm)	A–B	Area underneath curve

**Table 3 nutrients-14-04224-t003:** Micro-architectural parameters of right femur cortical bone (Ct.Th & Ct.Ar/Tt.Ar) and L5 trabecular bone (BV/TV, Conn.D, Tb.Th, & Tb.Sp) *.

Group	Ct.Th (mm)	Ct.Ar/Tt.Ar (%)	BV/TV (%)	Conn.D (mm^−3^)	Tb.Th (mm)	Tb.Sp (mm^−1^)
CON	0.315 ± 0.0211	56.0 ± 1.23	35.4 ± 2.43	35.0 ± 6.89	0.102 ± 0.00644	0.167 ± 0.0209
5-HS	0.329 ± 0.0278	55.4 ± 1.69	34.3 ± 1.63	40.3 ± 9.87	0.096 ± 0.00563	0.189 ± 0.0203
15-HS	0.345 ± 0.0288	56.6 ± 1.68	35.7 ± 1.99	40.0 ± 8.36	0.098 ± 0.00355	0.180 ± 0.0137

* Values represent mean ± SEM. Sample sizes for cortical parameters are *n* = 7 (CON), *n* = 8 (5-HS), *n* = 7 (15-HS). All trabecular parameters have a sample size of *n* = 8 per group.

**Table 4 nutrients-14-04224-t004:** Extrinsic mechanical properties of the right femur by diet group *.

Group	Stiffness (N/mm)	Yield Load (N)	Post-Yield Displacement (mm)	Work-to-Fracture (N * mm)	Maximum Load (N)
CON	67.40 ± 5.427	14.27 ± 0.934	0.1227 ± 0.02296	3.995 ± 0.4750	17.83 ± 0.8870 ^A^
5-HS	68.35 ± 6.657	13.69 ± 1.343	0.1731 ± 0.05001	4.318 ± 1.116	17.21 ± 1.251 ^A,B^
15-HS	54.85 ± 4.395	11.63 ± 1.054	0.1397 ± 0.02212	3.396 ± 0.4353	14.16 ± 0.5462 ^B^

* Values represent mean ± SEM. The sample size for each group is *n* = 8. Where significance was found, post hoc analysis was performed, and group comparisons are indicated by letter superscripts. Groups sharing the same superscript within columns are not significantly different (*p* > 0.05) whereas groups with different superscripts are significantly different (*p* < 0.05) per the *Tukey–Kramer* test.

## Data Availability

The data presented in this study are available on request from the corresponding authors.

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
