# Peer review of "Dietary Hempseed Decreases Femur Maximum Load in a Young Female C57BL/6 Mouse Model but Does Not Influence Bone Mineral Density or Micro-Architecture"

_nutrients, 2022, doi:10.3390/nu14204224_

Round 1

Reviewer 1 Report

Dear Authors, 

I have read carefully the manuscript entitled “Dietary Hempseed Decreases Femur Maximum Load in a Young Female C57BL/6 Mouse Model but Does Not Influence Bone Mineral Density or Micro-Architecture” submitted to Nutrients as an original article.

L39 I suggest removing presented unit of BMD form this sentence. In general, bone mineral density should be expressed in g per unit of volume, however DXA allows only to express BMD as a densitometric planar projection on surface. Therefore, in DXA measurements BMD values are given as g/cm2.

L58 “skeletal bone”  consider rephrasing. While phase  "Spinal and Skeletal Bone" (incorrect in my opinion) can be sometime found, in your study you not only analyse long bone (femur), but also   vertebra.

L65 and others : Please consider changing phase “work-to -failure” to “work-to-fracture”

L75 and others: the abbreviation “DXA” is recommended by the International Society for Clinical Densitometry 

L81: 79 and others : Please change “chickens” to “laying hens” . In animal sciences term “chickens” usually means “broiler chickens” and as you know there is a huge difference in the effects of feed additives on bone properties between young broiler chickens (up to 7wks of age) and mature laying hens (starting from 18wks of age). Please see also my comments below.

L112 AIN-93G

L137, L135 and fig 1. The shortcut in the text resulted in an awkward misunderstanding of the context by the author of Figure 1. The correct name is “femoral neck” and “femoral head”, and as you wrote “femoral head” was excluded from the analysis. In figure 1 shows as excluded region … mouse head. Please correct the description and, of course, the figure.

L223 and L224 Please use SI units

L271 Please add the information that interaction was included in the model

L289 was this decrease analyzed for statistical significance ?

Table 3: how uCT apparatus was calibrated? Genially for bone, HU for trabecular bone is below 500 HU and for cortical bone below 2000 HU, assuming that the calibration was performed air as -100 HU and for water as +100 HU.  please provide necessary details or references in M&M section

L328 lower

L331-334 were your data checked for normal distribution ? Similarly, was the equality of variances verified? failure to meet ANOVA assumptions may be the reason for the lack of significance between treatments as ANOVA requires normally distributed data.

L380-382 Combine with similar sentence in L 403.

L419 fracture load

L420 The same as in L81-91. In [23] a 52 wks old laying hens were used, not “chickens”. Laying hens at the age of 52wks are after their first laying period, when they suffer for extreme bone loss and osteoporosis. In your study, you have used young , growing mice. Therefore, the age might be the main factor influencing your results as there might be a huge difference in bone turnover rate between  young growing mice and old bird with osteoporotic bones.

L432-443. As you know, livestock animals, especially monogastric, are generally supplemented with enzyme cocktail (phytase, xylanase, and others) to overcome the antinutritional factors present in grains and increasing the bioavailability of phytate-bound phosphorus. enzyme supplementation might be an additional factor influencing the observed differences between your study and reported in [23]. However, the information about potential enzyme supplementation in [23] is missing and therefore this hypothesis cannot by verified.

Reviewer 2 Report

A number of human and animal studies have shown that phytic acid reduces the bioavailability of essential minerals, such as calcium (see suggested references). In the present study, it looks like the phytic acid content of hempseed does impede bone development, as shown by the interaction between age and diet (p. 8).

Is rodent chow a suitable approximation of a normal human diet? This point is important because the authors are extrapolating from lab mice to humans. Since rodent chow has a high cereal content, it must also be high in phytic acids. If the authors had used rodent chow with zero phytic acid content, would the control group have differed even more from the high hempseed diet group in terms of bone development?

Corrections

Line 41 – replace “painful, costly” with “painful and costly” (alternatively, you could add another “can be” before “costly”)

Line 69 – the word “both” cannot be used with “and/or.” Either drop “both” or replace “and/or” with “and”

Line 81 – replace “insight regarding” with “insight into”

Line 161 – insert “the” before “femur”

Line 178 – insert “the” before “L5 vertebra”

Line 181 – insert “the” between “excluded” and “cortical shell”

Line 182 – insert “the” between “only” and “trabecular”

Line 199 – replace “were” with “was”

Line 200 – replace “value” with “values”

Line 201 – replace “the mean gray value” with “that averaged value”

Lines 201 and 203 – replace “vertebrae” with “vertebra”

Line 230 – replace “lowering” with “lowered”

Line 275 – insert a comma after “interest”

Line 356 – “L5 vertebrae of each mouse” – How can a single mouse have more than one L5 vertebra? (vertebrae is plural)

Line 358 – see above

Suggested references

Harrison, D.C.; Mellanby, E. Phytic acid and the rickets-producing action of cereals. Biochem. J. 1939, 33, 1660-1680. https://doi.org/10.1042%2Fbj0331660

McCance, R.A.; Widdowson, E.M. Mineral metabolism of healthy adults on white and brown bread dietaries. J. Physiol. 1942, 101, 44-85. https://doi.org/10.1113%2Fjphysiol.1942.sp003967

McCance, R.A.; Widdowson, E.M. Mineral metabolism on dephytinized bread. J. Physiol. 1942, 101, 304-313. https://doi.org/10.1113%2Fjphysiol.1942.sp003984

Wills, M.R.; Phillips, J.B.; Day, R.C.; Bateman, E.C. Phytic acid and nutritional rickets in immigrants. Lancet. 1972, 299(7754), 771-773. https://doi.org/10.1016/S0140-6736(72)90523-5

Round 2

Reviewer 1 Report

Dear Authors,

Thank you for corrections made. However, I still think that figure 1 needs modifications. The results of whole body DXA are not presented in the manuscript. There is no need to emphasize a fact that mouse head was excluded form analysis.  Other skeletal bones, such as the humerus, collarbone, and tibia, were also not scanned. L134-136 read: “To obtain the BMD values for specific bones, the region of interest (ROI) was adjusted to contain only the bone of interest.”  Please replace Figure 1 with one showing only ROIs for L5 and femur and remove L146-148 – the information that adianta manuscript is under preparation is unnecessary, as it does not bring any knowledge to the presented results and discussion.